# Cell Adhesion at the Tight Junctions: New Aspects and New Functions

**DOI:** 10.3390/cells12232701

**Published:** 2023-11-24

**Authors:** Nicolina Wibbe, Klaus Ebnet

**Affiliations:** 1Institute-Associated Research Group “Cell Adhesion and Cell Polarity”, Institute of Medical Biochemistry, ZMBE, University of Münster, Von-Esmarch-Str. 56, D-48149 Münster, Germany; 2Cells-in-Motion Cluster of Excellence (EXC 1003-CiM), University of Münster, D-48419 Münster, Germany

**Keywords:** cell–cell adhesion, force sensing, junctional adhesion molecule, JAM-A, phase separation, tight junction

## Abstract

Tight junctions (TJ) are cell–cell adhesive structures that define the permeability of barrier-forming epithelia and endothelia. In contrast to this seemingly static function, TJs display a surprisingly high molecular complexity and unexpected dynamic regulation, which allows the TJs to maintain a barrier in the presence of physiological forces and in response to perturbations. Cell–cell adhesion receptors play key roles during the dynamic regulation of TJs. They connect individual cells within cellular sheets and link sites of cell–cell contacts to the underlying actin cytoskeleton. Recent findings support the roles of adhesion receptors in transmitting mechanical forces and promoting phase separation. In this review, we discuss the newly discovered functions of cell adhesion receptors localized at the TJs and their role in the regulation of the barrier function.

## 1. Introduction

The intercellular junctions of epithelial and endothelial cells are specializations of the cell membrane that physically integrate individual cells into tissues. These junctions consist of distinct substructures visible via ultrastructural analysis, such as tight junctions (TJs), adherens junctions (AJs), or desmosomes, with different functions at the level of the individual cell, of the organ, and of the entire organism, which, for example, include apical-basal polarity, barrier formation, resistance to the mechanical strain and propagation of mechanical forces, and morphogenesis [1,2,3,4,5]. The integrity of these junctions is necessary for the maintenance of functional tissues. Abnormalities in the organization or function of the junctions are frequently associated with disorders like inflammation and malignant transformation [6,7,8]. Consistent with the necessity to develop adhesive systems as a prerequisite for the development of multicellular organisms, the origination of proteins mediating cell–cell adhesion dates back to the closest extant unicellular relative of multicellular organisms [9,10]. Interestingly, while AJs are used to mediate adhesion in the earliest metazoa, both desmosomes and TJs exist only in vertebrates [11,12]. Desmosomes have probably evolved from AJs as they use similar molecular components to link adjacent cells [12], whereas TJs have probably evolved from the Septate Junctions (SJs) of invertebrates, which are morphologically different from TJs but are functionally homologous to TJs and contain molecular components that are related to the components found in vertebrate TJs [13].

All three types of junctions are composed of several multiprotein complexes. These complexes consist of at least one integral membrane protein that interacts with a cytoplasmic protein which, together with other scaffolding proteins, forms a junctional plaque that is linked to the actin cytoskeleton, the intermediate filament system, or the microtubule system [14,15,16]. In addition to their role in connecting the cellular filamental systems to the cell–cell junctions, all three types of cell junctions serve as hubs for signaling events that regulate proliferation, differentiation, or cell migration [17,18,19]. The integral membrane proteins localized at the different types of junctions thus not only physically link adjacent cells to form a structural continuum of epithelial cells, but also regulate the specific localization of signaling hubs along the intercellular junctions.

Interestingly, the most diverse repertoire of adhesion molecules and integral membrane proteins is localized at the TJs. As opposed to AJs and desmosomes, which use cadherins and members of the nectin family of the immunoglobulin superfamily (IgSF) to interact with adjacent cells, TJs make use of various members of the IgSF, members of the claudin family, members of the TJ-associated Marvel protein (TAMP) family, and members of the Crumbs family of integral membrane proteins to regulate their function. In this review article, we provide a brief overview on the emerging new functions of TJ-associated integral membrane proteins. The interested reader is referred to comprehensive recent reviews on the structural organization of TJs and the functional roles of integral membrane proteins in TJ formation [20,21,22,23,24,25].

## 2. Integral Membrane Proteins at the TJs

Integral membrane proteins localized at the TJs can be classified into four major groups: Crumbs homolog 3 (CRB3), a member of the Crumbs family of proteins; claudins; TAMPs; and members of the IgSF of adhesion molecules (Figure 1).

### 2.1. CRB3

CRB3 is a member of the vertebrate Crumbs family of proteins, which are homologous to Drosophila Crumbs. CRB3 and has a very short extracellular region that consists of 33 amino acids (AA), a single transmembrane region, and a short cytoplasmic region of 40 AA [26]. In addition to its localization at TJs, CRB3 is also localized at the apical membrane domain of epithelial cells [27]. As opposed to Drosophila Crumbs, the vertebrate CRB3 isoform localized at the TJs is most likely not involved in homophilic or heterophilic interactions [28].

### 2.2. Claudins

Claudins are a family of tetraspan transmembrane proteins comprising 27 members [29,30]. Notably, claudins support cell aggregation when expressed in fibroblasts, indicating that their adhesive activity not only regulates their localization at homotypic cell–cell contacts but also contributes to the physical cell–cell adhesion at the TJs [31]. A central property of claudins is their ability to multimerize by interacting both in cis and in trans with either the same or a different claudin family member to form strands that are visible via freeze-fracture electron microscopy (EM) [32,33]. Claudin-based strands can act as occluding barriers for water and small solutes, as well as anion- or cation-selective paracellular channels, and are the principal paracellular permeability regulators of epithelial and endothelial barriers [34].

### 2.3. TAMPs

Members of the TAMP family include occludin, tricellulin/MarvelD2, and MarvelD3 [35]. Similar to claudins, TAMPs are tetraspan transmembrane proteins, but based on their sequence homologies, they are not related to claudins. Their characteristic feature is a conserved four-transmembrane “MAL and related proteins for vesicle trafficking and membrane link” (Marvel) domain [36]. Of the three TAMPs, tricellulin is unique as it is enriched at sites of contact between three cells (tricellular TJs, tTJs) [37]. Heterotypic interactions between tricellulin and MarvelD3, as well as between occludin and MarvelD3, have been described via co-immunoprecipitation (CoIP) and via Förster resonance energy transfer (FRET) experiments. These interactions most likely occur in cis [35,38]. A homophilic interaction in trans has been found for occludin but not for tricellulin or MarvelD3 [38,39]. While several studies have suggested that TAMPs per se are not essential for the development of an epithelial barrier function [40,41,42], recent findings indicate that tricellulin is required for the establishment of the barrier function in mammary gland-derived epithelial cells [43].

### 2.4. IgSF Proteins

IgSF members localized at the TJs include the junctional adhesion molecule (JAM) family members JAM-A and JAM-C [44], and the JAM-related adhesion molecules Coxsackie- and Adenovirus-Receptor (CAR), JAM4, CAR-like membrane protein (CLMP), and Endothelial Cell-Selective Adhesion Molecule (ESAM, in endothelial cells) [45,46,47,48]. All these IgSF proteins can undergo trans-homophilic interactions which stabilize their localization at cell–cell contacts. In addition, the trans-homophilic activities of CAR, JAM4, CLMP, and ESAM support cell aggregation after transfection in cells [45,46,47,49], suggesting that their adhesive activities contribute to the strength of the physical interaction at the TJs. Angulins (Angulin-1, -2, -3) are IgSF proteins with a single N-terminal Ig-like domain [50]. Among all other IgSF proteins localized at the TJs, angulins are unique in that they are enriched at the tTJs [51]. A main function of all three angulins is to recruit tricellulin to the tTJs [51,52]. The ectopic expression of angulin-1 in L cell fibroblasts resulted in angulin-1 enrichment at cell–cell contacts, suggesting that angulin-1 is engaged in homophilic or heterophilic trans interactions [52].

Strikingly, many of the integral membrane proteins localized at the TJs contain a C-terminal PDZ domain-binding motif (PBM), through which they can interact with the PDZ domains present in many TJ-localized scaffolding proteins (Table 1).

The presence of multiple PDZ domains in many scaffolding proteins localized at the TJs (Figure 1), combined with the promiscuity of PDZ domains in ligand binding, allows for the incorporation of several integral membrane proteins in a single scaffolding protein-organized complex. Vice versa, individual membrane proteins can recruit and assemble distinct protein complexes at the TJ. These biochemical properties may, in part, explain the high complexity and dynamics of protein complexes at the TJs [27].

## 3. Integral Membrane Proteins as Anchors for Cytoskeletal Elements at the TJs

Similar to AJs and desmosomes, the TJs are connected to cytoskeletal elements, in particular to the actomyosin network, but also to microtubules (MT) and intermediate filaments (IF). These connections are important for many biological functions, including cell division, apical-basal polarity, the transmission of mechanical forces, and many more. All three types of elements form filamentous networks beneath the apical membrane domain of polarized epithelial cells [54]. The anchoring of cytoskeletal filaments at the TJs is mediated by cytoplasmic scaffolding and adapter proteins associated with the integral membrane proteins at the TJ. The most prominent scaffolds for cytoskeletal elements at the TJs are the zonula occludens (ZO) proteins ZO-1, ZO-2, and ZO-3, which all interact with F-actin, and the cingulin family proteins cingulin (CGN) and CGN-like1 (CGNL1)/junction-associated coiled-coil protein (JACOP), which interact with both F-actin and microtubules and, additionally, with ZO proteins [55]. Of note, several integral membrane proteins directly interact with ZO proteins (Figure 1) suggesting that actin filaments can be tethered to TJs by several membrane proteins. Also, CGN exists in a complex with JAM-A [56] and with ZO-1/2 [57], suggesting that microtubules can be similarly tethered to TZJs through multiple mechanisms [55]. A recent study provided evidence for the interaction of the intermediate filament protein keratin 76 with claudin-1 [58]. Although it is not clear whether this interaction is direct or indirect, this observation provides the first clue for a potential mechanism for the tethering of intermediate filaments at the TJs. For more detailed descriptions of the functional roles of cytoskeletal element anchoring at the TJs, the interested reader is referred to recent reviews [15,59,60,61].

## 4. Integral Membrane Proteins and the Paracellular Barrier Function of TJs

A principal function of TJs is the establishment of a selective paracellular barrier to the free diffusion of water and solutes. This function is mainly accomplished by claudins [29]. Claudins are the molecular basis for TJ strands and the determinants of the pore permeability pathway of TJs, which are characterized by a size- and charge-selective permeability for molecules with a hydrodynamic radius of approximately 4–6 Å [62,63]. The expression of a specific set of claudins enables the formation of ion-selective or barrier-forming channels, which provides a mechanism for adaptation to the needs of a given tissue in an organism [25,64].

However, the model of the claudin-based regulation of paracellular permeability only accounts for the pore permeability pathway. The second major permeability pathway regulated by TJs, the leak pathway, which is not charge-selective and which allows a limited flux of large molecules with a hydrodynamic radius of up to approximately 120 Å [62,63], is not dependent on claudins. MDCK cells lacking all claudins still develop close membrane appositions and form a barrier to 150 kDa FITC-dextran molecules which have a radius of approximately 90 Å [65]. This barrier is lost upon the additional depletion of JAM-A. Interestingly, close membrane appositions observed in the absence of claudins are strongly reduced, albeit not completely absent, in MDCK cells lacking claudins and JAM-A, suggesting that JAM-A clustering as a result of trans-homophilic interactions [66] forms a barrier to the diffusion of macromolecules. In line with this assumption, the ectopic expression of JAM-A in CHO cells, which express neither claudins nor JAM-A, and which lack TJ strands, results in an increased barrier for 40 kDa FITC-dextran (with a hydrodynamic radius approximately 66 Å) [67].

The development of barrier-forming epithelial monolayers requires the formation of a continuous TJ strand network that seals the entire epithelium. The sealing of cell vertices, the common points of three or more neighboring cells, creates a challenge to this requirement. Early studies using freeze-fracture EM showed that, at the sites of tTJs, the paired strands of two neighboring cells develop vertical extensions, so-called central sealing elements, which are cross-bridged with the central sealing elements of the two other bicellular contact sites to form a central tube [68,69]. Tricellulin has been identified as a structural and functional component of tTJs [37] (Figure 2). Tricellulin is enriched at tTJs, and its depletion results in discontinuities of the TJ strand network along bicellular TJs (bTJs) and in a loss of the epithelial barrier function [37], strongly suggesting that tricellulin links the bTJ strand network to the tTJs to warrant continuity of the network at the interfaces of bTJs and tTJs. While this model could not explain how cells prevent the leakage of solutes through the central tube that is lined by the three central sealing elements in the center of a tTJ (Figure 2), new evidence provides a possible explanation. Tricellulin is linked to the actomyosin network via α-catenin and vinculin [43], two adapter proteins known to link cadherins to the actin cytoskeleton at AJs. Through its interaction with α-catenin and vinculin, tricellulin recruits actin filaments to the central sealing elements present at the tTJs. The myosin II motor-driven contraction of antiparallel actin filaments associated with tricellulin generates a close apposition of the central sealing elements originating from the three cells to minimize the intercellular space at tTJs [43].

Interestingly, the function of tricellulin is closely linked to the functional activity of angulin-1. As mentioned before, angulin-1 is localized specifically at tTJs [52]. The depletion of angulin-1 phenocopies the depletion of tricellulin, with discontinuities of occludin localization at bTJs, a widening of the intercellular space at tTJs, and impaired paracellular barrier formation [43,52]. While the depletion of angulin-1 results in a loss of tricellulin from tTJs, angulin-1 retains its specific localization at TJs after the depletion of tricellulin [43,52], indicating that one major function of angulin-1 is the recruitment of tricellulin to tTJs. Angulin-1 has functions that go beyond the mere recruitment of tricellulin to tTJs. Angulin-1 recruits ZO-1 and, via ZO-1, claudin-2 to the more basal region underneath the typical TJs at tricellular contacts, which most likely represent the central sealing elements formed by vertical TJ strands [42]. In addition, Angulin-1 is responsible for the close membrane apposition of the three cells at tricellular contact sites, and, surprisingly, not only at the level of the tTJs but also at the level of the desmosomes [42]. This function of angulin-1 is independent of JAM-A or claudins. Interestingly, in MDCKII cells used in the study by Sugawara and colleagues [42], tricellulin appears to be required for the connection of TJ strands to the central sealing elements, but not for the barrier for 332 Da fluorescein, which suggests that the function of angulin-1 in forming a barrier for small uncharged solutes is independent of tricellulin. Together with the observation in mammary gland-derived Eph4 cells [43], these observations also indicate that the specific contribution of a given integral membrane protein to the barrier function might depend on the given tissue.

## 5. Integral Membrane Proteins and the Mechanical Force Load on TJs

Epithelia are exposed to mechanical forces that originate both from external sources (extrinsic forces) and from epithelial cells themselves when sites of adhesion are coupled to the contractile actomyosin cytoskeleton (intrinsic forces) [70,71]. During the last decade, the mechanisms through which forces are sensed and propagated between cells have been extensively characterized, resulting in the idea that cell–cell adhesion receptors connected to the actomyosin cytoskeleton through adapter proteins link cells and thus can generate tissue-scale tension. In addition, through the tension-dependent conformational changes of adapter proteins, alterations in mechanical forces can be sensed and translated into intracellular signaling pathways [4,70,72]. While this concept has been elaborated in detail for AJs [73], the sensing of tensile forces and their transduction at the TJs is now beginning to be understood in more detail [74].

As part of the apical junctional complex, TJs are frequently subjected to mechanical challenges evoked by cell shape changes as they occur during cell extrusion or cytokinesis, as well as those generated during morphogenetic movements. Resisting these mechanical forces is particularly challenging as breaches in TJ strands could impair the barrier function [75], possibly resulting in a loss of the organ-specific absorptive function of the epithelium, or in inflammation. The recent observations that ZO-1 and ZO-2 undergo actomyosin-dependent conformational changes, and that ZO-1 is under tensile stress, have provided the first direct evidence that mechanisms of force-sensing and -translation analogous to those described in AJs operate at the TJs [76,77]. When connected to the actomyosin cytoskeleton, both ZO-1 and ZO-2 adopt an open conformation that exposes a region of approx. 400 AA required for the binding of occludin and the transcription factor DNA-binding protein A (DbpA) [76]. This open conformation is required to recruit occludin and DbpA to cell–cell junctions and to mediate occludin- and DbpA-dependent functions [76]. Recent findings have further confirmed that several functions of ZO-1 in polarized epithelial cells, including junction formation and cell morphology, are dependent on actomyosin-mediated forces, and that ZO-1, surprisingly, also regulates tension on AJs and traction forces on the growth substrate, indicating a mutual regulation of cytoskeletal tension and ZO-1 function [78,79].

Interestingly, the tensile stress acting on ZO-1 is regulated by JAM-A. JAM-A directly interacts with the PDZ3 of ZO-1 [56,80,81] and exists in a common complex with ZO-1 and also ZO-2 [82] (Figure 3). Similar to the depletion of ZO-1, the depletion of JAM-A increases RhoA and actomyosin activity, which is mediated by the recruitment of p114RhoGEF to cell–cell junctions [77]. At the same time, JAM-A depletion increases the tension on ZO-1, which is most likely mediated by increased actomyosin activity. By regulating the local activity of RhoA, JAM-A has a direct impact on the mechanical load acting on ZO-1 at TJs. It is also noteworthy that apart from the regulation of this intrinsic mechanical tension acting on TJs, JAM-A may also regulate the extrinsic force load on cells. The direct application of pulling forces on JAM-A through trans-homophilic JAM-A interaction activates various RhoA GEFs and increases RhoA activity [83]. Also, recent experiments with recombinant proteins indicate that the strength of the JAM-A–ZO-1 interaction is modulated by mechanical forces that act on this binary interaction [84]. Besides ZO-1, JAM-A interacts with the AJ-localized protein afadin [80]. Despite comparable affinities, the interaction of JAM-A with the PDZ domain of afadin is reinforced in response to the mechanical load over a given force range, whereas, by contrast, the interaction of JAM-A with the PDZ3 of ZO-1 is destabilized in response to a mechanical load [84]. These observations thus indicate that the protein interaction networks at TJs dynamically change in the presence of mechanical forces. In addition, since JAM-A is not exclusively localized at the TJs [85,86], mechanical forces may also shift the balance between TJ- and AJ-associated protein complexes [87].

## 6. Integral Membrane Proteins and Phase Separation at the TJs

The TJs represent an adhesive structure at the apical region of cell–cell junctions that is highly enriched in cell adhesion molecules, cytoplasmic adapter proteins, and signaling proteins [2,88]. As mentioned before, many of the adapter proteins are modular in nature and contain multiple protein–protein interaction domains such as PDZ domains, SH2 domains, or SH3 domains [89], which promotes the formation of the multimolecular protein complexes involved in the initiation of signaling cascades triggered by cell–cell adhesion. Studies, in the recent years, have shown that the formation of such large protein complexes is facilitated by a process called liquid–liquid phase separation (LLPS) [90,91].

LLPS is driven by weak interactions between multivalent molecules which, at a specific concentration, start to exclude the surrounding solution and form aggregates. This process promotes the formation of molecular condensates in the cytoplasm and in the nucleus (three-dimensional (3D)-LLPS), or the assembly of clusters consisting of adhesion receptors and their cytoplasmic binding partners in the plasma membrane (2D-LLPS, membrane-associated LLPS) [91]. At the plasma membrane, cluster formation can be triggered by the phosphorylation of transmembrane receptors, which provides a means for the dynamic and reversible formation of protein clusters [92]. In addition, transmembrane receptors frequently dimerize, either constitutively or in response to ligand binding [92], enabling a dimer to interact with a distinct multivalent cytoplasmic scaffolding protein. This property contributes to a positive feedback mechanism for the clustering of transmembrane receptors [93], a mechanism that might be of particular importance for the “leak pathway” of TJs [62].

Recent evidence indicates that the LLPS-mediated condensation of the three zonula occludens proteins ZO-1, ZO-2, and ZO-3, drives the accumulation of TJ components, including claudins, at a specific membrane compartment, and that this separation of ZO proteins into membrane-attached compartments is necessary for the formation of claudin-based strands and, thus, for the development of functional TJs, both in cell culture and in the organism [94,95,96]. Notably, the sequestration of ZO proteins from cytoplasmic condensates into membrane-associated compartments is dependent on mechanical forces [95,96], suggesting that the force-mediated conformational changes of ZO proteins drive the formation of multiprotein complexes at the TJs. As recently shown, ZO-1 is under tensile stress [77], which induces an open conformation that exposes binding sites for interaction partners, including occludin [76]. The direct interaction of JAM-A with the PDZ3 domain of ZO-1 suggests that JAM-A at the TJs may act as a membrane receptor for ZO-1, which triggers the transition of ZO-1 cytoplasmic condensates into the ZO-1 surface condensates required for TJ formation. In vitro partition assays revealed a co-clustering of JAM-A with claudin-2 [97]. Through its interaction with ZO-1 and ZO-2, JAM-A appears to be a likely transmembrane component at the TJs that initiates the formation of surface condensates. At the same time, JAM-A’s clustering at the TJs may be the result ZO protein condensate formation at the TJs.

## 7. Lipid Modifications of Integral Membrane Proteins at TJs

TJs undergo continuous remodeling not only in response to mechanical challenges but also at steady state [98,99,100]. This dynamic behavior is also reflected in the constant reorganization of TJ strands [98], accompanied by a rapid diffusion of TJ components within the membrane and a high exchange rate between the membrane and intracellular pools of some other TJ components [99]. A rapid turnover of proteins can be regulated by reversible post-translational modifications, including the attachment of phosphate residues, the addition of ubiquitin and ubiquitin-related residues, or the addition of acyl chains. In fact, many TJ-localized membrane proteins are post-translationally modified [101,102].

S-palmitoylation, i.e., the reversible addition of a saturated C16 fatty acid chain to specific cystein residues via a thioester bond [103], appears to be a mechanism underlying the specific localization of several membrane proteins at the TJs. The S-palmitoylation of membrane proteins promotes their association with membrane domains that are enriched for cholesterol and sphingolipids, so-called lipid rafts [104,105]. Of note, TJs have been described as raft-like membrane microdomains that are enriched in cholesterol and sphingolipids [106,107]. S-palmitoylation has been identified in several members of the claudin family [108,109,110,111,112,113], as well as in JAM-C [114] and Angulin-1 [115]. The S-palmitoylation of these TJ membrane proteins may thus contribute to the specific and perhaps dynamically regulated localization at the TJs.

The role of S-palmitoylation has recently been highlighted by studies showing that claudins which lack the PBM and thus cannot be recruited by ZO proteins [116,117] still can localize to TJs and form TJ strands, whereas a palmitoylation-deficient claudin mutant is no longer localized at the TJs [113]. One study revealed that ZO proteins, in addition to their widely accepted function in acting as a scaffold for a number of integral membrane proteins, including claudins, IgSF members, and occludin, at the TJs, also regulate the accumulation of cholesterol at the TJs through a mechanism that involves the formation of the circumferential actin ring at the TJs [113]. Importantly, this latter function of ZO proteins is sufficient to promote the formation of claudin-dependent TJ strands. This study, thus, also established a hierarchy of ZO protein-mediated functions in TJ formation and provided important new insights into the functional relevance of the palmityolation of TJ membrane proteins.

## 8. Summary and Conclusions

The TJs maintain a selective permeable seal between epithelial cells to establish a barrier between tissue compartments. Live cell imaging experiments using fluorescently tagged TJ proteins have revealed that TJ strands are highly dynamic structures at steady state. The dynamic properties of TJ strands are most likely an evolutionary adaptation to the mechanical forces imposed on cell–cell junctions during cellular processes like cell division or cell extrusion, but also during developmental processes such as cell rearrangements and tissue stretching and bending. A dynamic reorganization of TJ strands allows the TJs to maintain an intact barrier in the presence of mechanical challenges. Cell adhesion receptors at the TJs are critical in sensing and relaying mechanical challenges at the cellular scale. The association of several different adhesive membrane proteins with the force-sensor and actin-interactor ZO-1 provides a mechanism as to how forces are sensed and translated into intracellular signaling cascades by cell adhesion receptors. Also, the recent identification of cytoplasmic ZO-1 condensates and their transition into surface condensates, presumably triggered by ZO-1 binding to JAM-A, provides a possible mechanism for a dynamic regulation of the protein composition at the TJs. A better understanding of the role of cell adhesion receptors and their interaction with scaffolding proteins in TJ physiology will be a challenging task for future studies.

## Figures and Tables

**Figure 1 cells-12-02701-f001:**
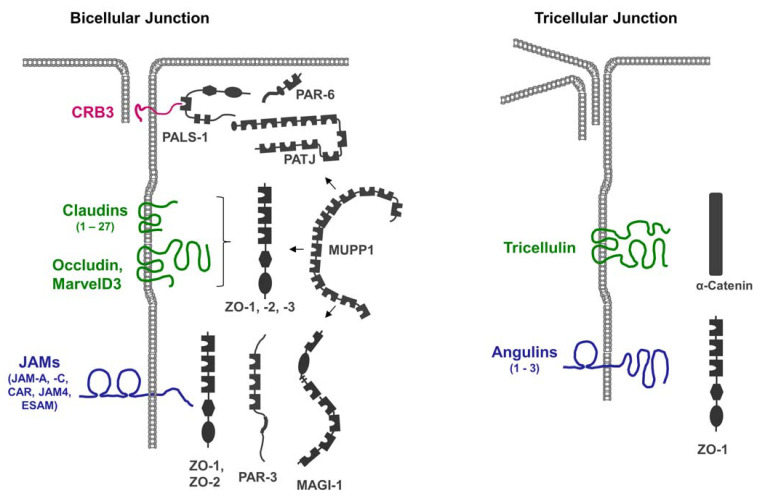
Integral membrane proteins at TJs and their interaction with scaffolding proteins. Tight junctions contain a number of membrane proteins including Crumbs3 (CRB3), claudins, TJ-associated marvel proteins (Occludin, MarvelD3, Tricellulin), and members of the immunoglobulin superfamily (JAMs, CAR, ESAM, Angulins). All membrane proteins at the TJ directly interact with one or several scaffolding proteins (depicted in grey color). Some scaffolding proteins interact with several integral membrane proteins, as well as with other scaffolding proteins. For example, MUPP1 can interact with JAM-A, CAR, and claudins, as well as with PALS-1, PAR-6, and ZO-3 (indicated by arrows). Tricellulin and angulins are specifically localized at tricellular TJs.

**Figure 2 cells-12-02701-f002:**
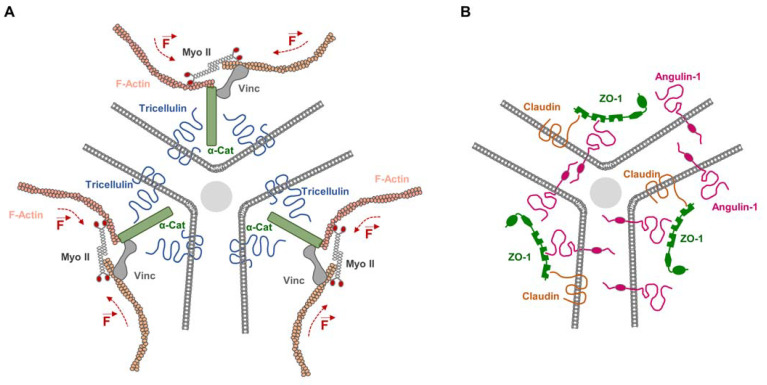
Membrane proteins at tricellular TJs and their association with cytoplasmic proteins. (**A**) Tricellulin is linked to the actomyosin network via α-catenin (α-Cat) and vinculin (Vinc). MyosinII-driven contraction of antiparallel actin filaments (indicated by broken line and force vector symbols) associated with tricellulin generates a close apposition of the central sealing elements originating from the three cells to minimize the intercellular space at tTJs. (**B**) Besides its role in recruiting tricellulin, angulin-1 also recruits ZO-1 and claudin-2 to the central sealing elements of the TJs, which are formed by vertical TJ strands (see text for details).

**Figure 3 cells-12-02701-f003:**
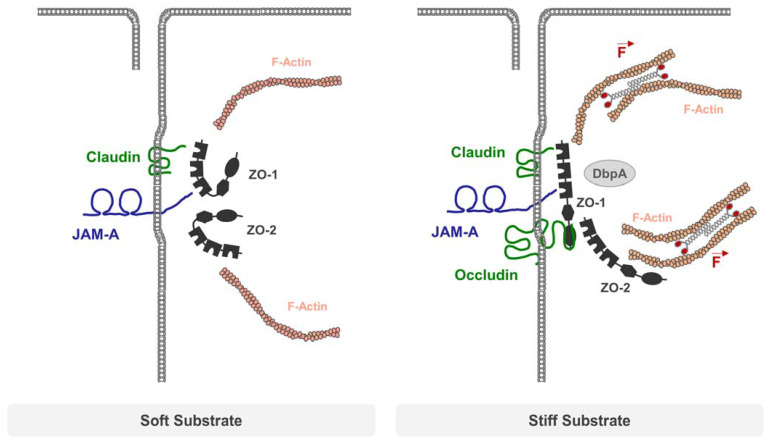
TJ regulation via mechanical forces. Soft substrates impose only weak forces on TJs, allowing ZO-1 and ZO-2 to adopt a closed conformation that does not interact with occludin. When grown on stiff substrates, the actomyosin-driven contractility of the actin cytoskeleton imposes mechanical forces on ZO-1, which adopts an open conformation. Open ZO-1 can recruit occludin and transcription factors like DbpA to regulate barrier function and gene expression. JAM-A controls actomyosin-based contractility at TJs by preventing the recruitment of p114RhoGEF to TJs through an unknown mechanism, thus limiting the mechanical load on TJs.

**Table 1 cells-12-02701-t001:** **PDZ domain binding motifs of integral membrane proteins at TJs**. The figure shows integral membrane proteins localized at the TJs. The total number of AA and the number of amino acids comprising the cytoplasmic regions are indicated. The five C-terminal AA comprising the PDZ domain binding motifs are also shown. A classification of PDZ domain binding motifs (PBM) is provided in reference [53]. The nomenclature applies to human proteins. AA are shown in single letter code. X refers to any AA. Abbreviations: PBM, PDZ domain binding motiv.

Integral Membrane Protein	Size (AA)	Cytoplasmic Region (AA)	COOH-Terminal Residues (PBM)
CRB3	120	40	- EERLI (+)
JAM-A	299	40	- SSFLV (+)
JAM-C	310	48	- SSFVI (+)
CAR	365	107	- DGSIV (+)
JAM4	407	122	- NTTVV (+)
ESAM	390	121	- AGSLV (+)
Angulin-1/LSR	649	369	- ESLVV (+)
Angulin-2/ILDR1	546	358	- RSVVI (+)
Angulin-3/ILDR2	639	432	- MSLVV (+)
Claudins (1–26)	207–305	27–66	- XXXYF (+)
Occludin	522	257	- DRQKT (−)
Tricellulin	558	196	- VQGYS (−)
MarvelD3	401	20	- EMFEF (+)

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
