# Peer review of "Cell Adhesion at the Tight Junctions: New Aspects and New Functions"

_cells, 2023, doi:10.3390/cells12232701_

Round 1
Reviewer 1 Report
Comments and Suggestions for Authors
Dear colleagues!
The paper is well written and is definitely prepared by highly competent professionals in the field.
I do have confidence that this submission addresses TJs in a relevant manner taking it as points of physiological regulation and cell-to-tissue transition that is a nee frontier is cell biology.
The manuscript summarizes recent and relevant data and rapid publication of this work is warranted.
Regards, Reviewer
Author Response
We thank this reviewer for his/her positive evaluation of our manuscript.
Reviewer 2 Report
Comments and Suggestions for Authors
This is a well-written, comprehensive, and timely review on tight junctions, their major components, and their overall structure and functions. The reviewed literature is overall well represented and up-to-date. The figures and tables are also nicely put together and help the reader follow the text. The review flows well and provides plenty of information for the expert in the field, but also simple descriptions of terms and concepts for the non-expert. I only have two suggestions: 1) since evolution of cell-cell adhesion complexes is mentioned in the Introduction, it would probably be nice to then clarify that tight junctions evolved only in vertebrates; 2) since PDZ domains and mechanotransduction are both discussed, I would suggest including discussion on a recent preprint: https://doi.org/10.1101/2023.09.24.559210
Author Response
As requested by the Reviewer, in the revised version of our manuscript we have mentioned that the TJs exist only in vertebrates, and we also mentioned that the Septate Junctions in invertebrates are the functional homologe and most likely the evolutionary ancestor of TJs. In addition, we discussed the recent preprint on the roles of the PDZ domains of Afadin and ZO-1 as mechanosensors. We thank the reviewer for pointing our attention to this article, which provides in fact important new aspects in the context of cell adhesion and mechanosensing at the tight junctions.
Reviewer 3 Report
Comments and Suggestions for Authors
The article is well written and comprehensive, but in some part not living up to its title. Although in Introduction it is mentioned that the cytoplasmic plaques of intracellular junctions, including TJs, are linked to cytoskeletal networks, this aspect is only marginally covered. The actomyosin system is addressed later on, but not the intermediate filament or microtubule systems. In light of reports showing that intermediate filaments and their associated proteins affect TJ integrity along with the actomyosin network, this aspects should be discussed.
Author Response
As requested by the Reviewer, we have discussed in more detail the intermediate filament (IF) system and the microtubule (MT) system. To stay within the scope of this review article, we have focussed the discussion on the function of cell adhesion receptors in anchoring IFs and MTs at the TJs. We have dedicated a separate paragraph to this topic.
Round 2
Reviewer 3 Report
Comments and Suggestions for Authors
The concerns raised have been addressed. Revision is improved.